# KHV: KVM-Based Heterogeneous Virtualization

Chunqiang Li [1,*], Ren Guo [2,*], Xianting Tian [2] and Huibin Wang [1]

1. Institute of VLSI Design, Zhejiang University, Hangzhou 310000, China
2. Alibaba Group, Hangzhou 310000, China
* Correspondence: chunqiangli@163.com (C.L.); guoren@kernel.org (R.G.)

**Abstract:** A KVM (Kernel-based Virtual Machine) is subject to the complexity of the Linux kernel and the difficulty and cost of safety certification; thus, it is not popularized in embedded high-reliability scenarios. This paper proposes a KVM-based Heterogeneous Virtualization (KHV), which is independent of hardware virtualization (KVM mandatory virtualization), follows the principle of static partitioning, localizes the hypervisor, and inherits the KVM software ecosystem. KHV balances the demands of static partitioning and flexible sharing in the embedded system. The paper implemented KHV on the RISC-V Xuantie C910 CPU-based SoC and conducted a performance comparison with KVM. The experiment shows that KHV is 50% smaller than KVM in terms of fluctuation, and KHV makes the guest OS have the same performance as the bare-metal OS in scheduler benchmarks, whereas KVM dropped an average of 28%.

**Keywords:** virtual machine; hypervisor; KVM; bare metal; RISC-V; physical memory protection; FV—full virtualization; PV—para virtualization; virtio; heterogeneous virtualization; static partitioning





## 1. Introduction

KVM is an open-source virtualization technology built in Linux and widely used in cloud computing because of its ecological advantages. After 14 years of development, it has built a rich software ecosystem and a large user group in cloud computing. It has become the mainstream virtualization solution in the data center. However, KVM cannot bring success in the data center to the embedded virtualization scenarios [1]. Compared with the requirements of data center virtualization scenarios for "massive clients, resources overcommit, dynamic virtual machine creation, destruction, and migration", embedded virtualization emphasizes non-interference between guests(ISO 26262 part6), static resources isolation, reliable performance, and high real-time performance. KVM does not meet the isolation requirements of embedded virtualization. At present, embedded virtualization scenarios are popular with hypervisors of Static Partitioning (e.g., Jailhouse [2], Bao [3], ACRN [4], Xen [5], Xvisor [6], QNX Hypervisor, Opensynergy COQOS Hypervisor), but they have problems such as lack of reliability, poor flexibility, and ecological closure.

This paper proposes a KVM-based Heterogeneous Virtualization (KHV) to overcome the difficulties of KVM deployment in embedded virtualization. KHV is a static partitioning technology for embedded scenarios and utilizes the KVM software ecosystem to share hardware resources flexibly. Regarding reliability, KVM is a hypervisor deeply embedded in the Linux kernel, which results in the reliability of the guest depending on the Linux kernel. Even if the guest RTOS reaches a high reliability level, the underlying Linux system still needs to pass higher reliability certification to meet the reliability requirements of the guest. It will significantly increase the complexity and cost of reliability certification. In terms of security, there have been many security vulnerabilities against the virtualization layer ([7–12]), and attackers exploit these vulnerabilities to attack hypervisors from guests, obtain sensitive data, and even obtain hypervisor administrator privileges. In addition, virtualization isolation cannot solve some security vulnerabilities caused by hardware

defects ([13–15]). KHV reduced the difficulty and cost of reliability certification and eliminated the security risks mentioned above by localizing and bypassing the underlying virtualization. Figure 1 shows the difference between KVM (Linux-based Hypervisor) and Xen (Embedded Hypervisor). This paper implements KHV with four instances described in Table 1.

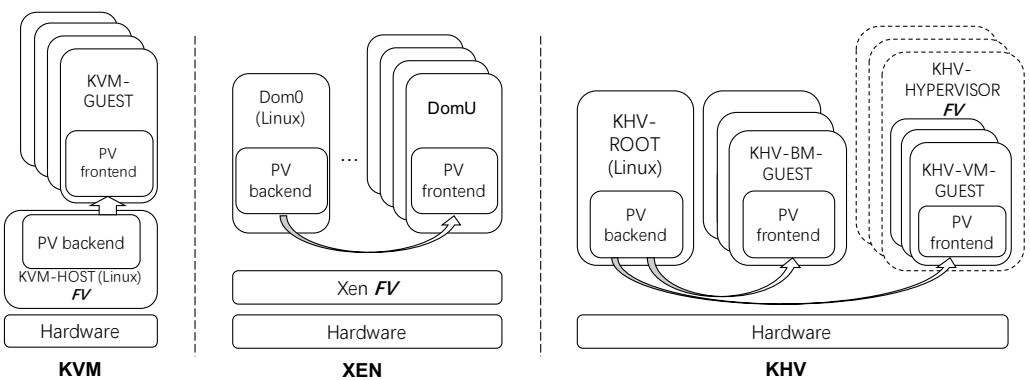

**Figure 1.** KVM vs. XEN vs. KHV.

**Table 1.** KHV instances noun explanation.

| Name | Description |
| --- | --- |
| KHV-ROOT | The root system, the first Linux operating system powered on, allocates system resources to create the remaining instances. |
| KHV-BM-GUEST | Bare-metal guest instances that do not rely on virtualization. |
| KHV-VM-GUEST | Virtualized guest instance. |
| KHV-HYPERVISOR | Passive Hypervisor for KHV-VM-GUEST, a bridge between KHV-ROOT and KHV-VM-GUEST. |

KHV-ROOT, KHV-BM-GUEST and KHV-VM-GUEST run at the same privilege level. So, KHV is a Nohype system, which eliminates KVM's poor isolation problem and the dependency on a low-level virtualization layer.

As shown in Figure 2, KHV-ROOT is the first root system booted by the SoC, which creates other KHV-BM/VM-GUESTs. KHV-BM-GUEST is a bare-metal guest that does not rely on CPU hardware virtualization and is an independent heterogeneous system once started. KHV-VM-GUEST is a virtual machine created by KHV-ROOT through KHV-HYPERVISOR using CPU hardware virtualization technology. KHV-HYPERVISOR is a lightweight virtualization layer that supports KHV-VM-GUEST to use virtual physical memory. It supports discrete physical memory through two-level address translation and memory sharing between guests. KHV-HYPERVISOR passively manages KHV-VM-GUEST without occupying CPUs and acts as a bridge for the KHV-ROOT backend service detailed in the "KHV Hypervisor" and "Virtio balloon" chapters.

KHV inherits the Para-Virtualization software ecosystem of KVM. The virtio-based communication technology is prevalent in data centers and embedded scenarios. Virtio uses shared memory and notification mechanism to provide data exchange services between OSes as the frontend and backend device driver (e.g., block, net, vsock). KHV inherits the virtio software ecology and user habits of KVM. Figure 3 shows that in the KHV system, KHV-ROOT provides virtio services for KHV-GUEST through the virtio bus.

This paper implements KHV on the RISC-V SoC platform, and KHV consists of a Linux kernel driver and a user mode management tool (kvmtool [16]). The motivation is introduced in the Related Work chapter. The implementation details are introduced in the Design chapter. KHV and KVM are tested, compared, and analyzed in the Evaluation chapter. In the Discussion section, we talk about future directions for improvement.

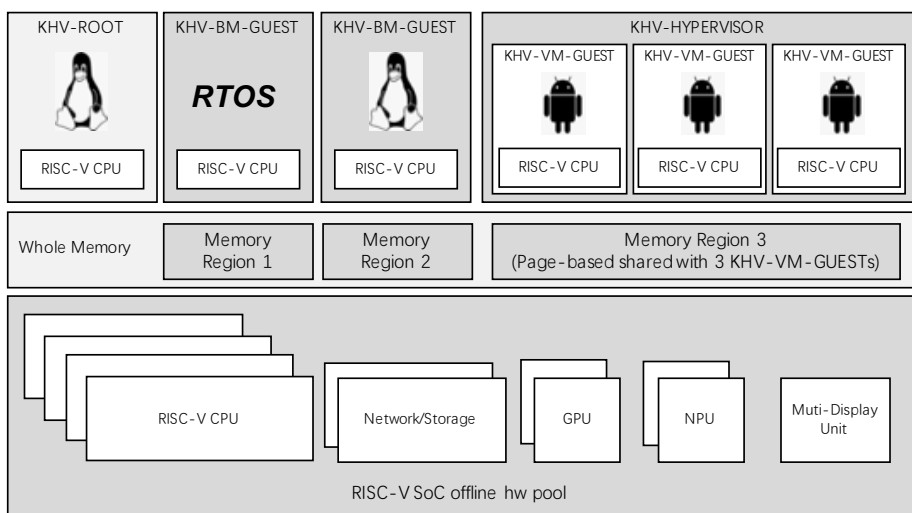

**Figure 2.** KHV on SoC.

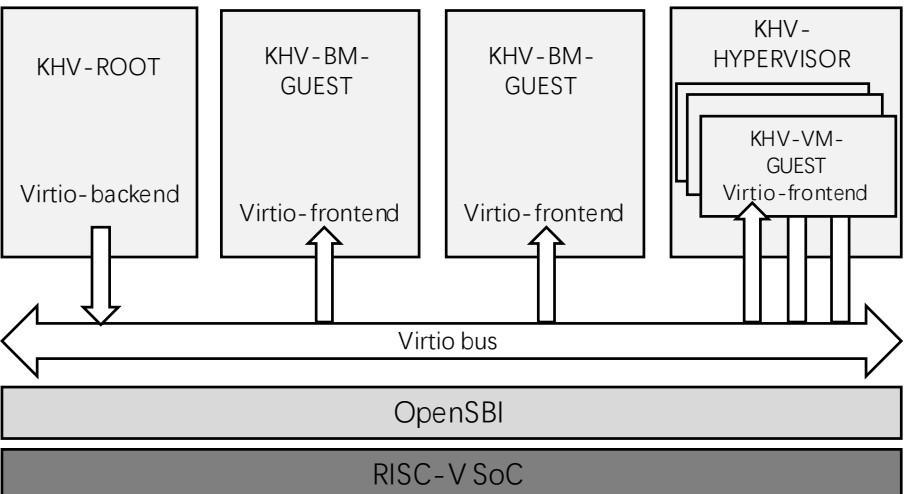

**Figure 3.** KHV virtio.

## 2. Related Work

Removal of the virtualization layer is not a fresh topic. It has been widely discussed and practiced in the literature, such as High-density Multi-tenant Bare-metal Cloud [17], NoHype in Cloud [18], and Look mum, no VM exits! (almost) [19]. They all attempt to eliminate the negative influence of virtualization tax on critical tasks for cloud computing, but none deal with embedded system scenarios. So, we innovate KHV, the bare-metal hypervisor solution for the embed system.

So far, there have been some existing bare-metal management solutions, such as Intel Many Integrated Core (MIC) Architecture, which is a heterogeneous scheme on Linux. However, its Linux backend virtio drivers are only test level and useless. Additionally, the MIC driver has been removed since Linux-5.10, for Intel does not produce MIC devices. In contrast, KHV reuses a mature and rich ecosystem of KVM.

## 3. Design

The KHV proposed in this paper uses the M-mode (Machine), HS-mode (Supervisor + Hypervisor), U-mode (User), VS-mode (Virtual Supervisor), and VU-mode (Virtual User) of the RISC-V architecture. The five runtime modes could meet the needs of various OSes. At the same time, KHV utilizes the Physical Memory Protection (PMP) natively provided by RISC-V and controls the access permission to the physical address range. PMP contains several entries (config with 8/16/32/64) to control the memory access of

KHV-GUEST. We can implement different types of hypervisors with RISC-V ISA. Next, We have a deeper understanding of the architectural design of KHV through the comparison with RISC-V KVM.

Figure 4 shows KVM running on RISC-V. The kvmtool runs in KVM-HOST and provides IO emulator and Para-Virtualization (PV) virtio backends service for KVM-GUEST. KVM-HOST (Linux) is responsible for running device drivers, kernel-mode PV backends (also known as vhost virtio backends), and Full Virtualization (FV) functionality. Other processes can also run PV backends (also known as vhost-user). There are two types of drivers running in KVM-GUEST, one are native IO drivers based on FV full virtualization (which is served by IO emulator), and the others are virtio-based PV frontends. Because the entire KVM system relies on KVM Host(Linux) as the base hypervisor, once there is a problem with Linux, it also affects the upper-layer KVM-GUEST, which violates the design principle of ISO-26262 part6 that does not interfere with each other between systems.

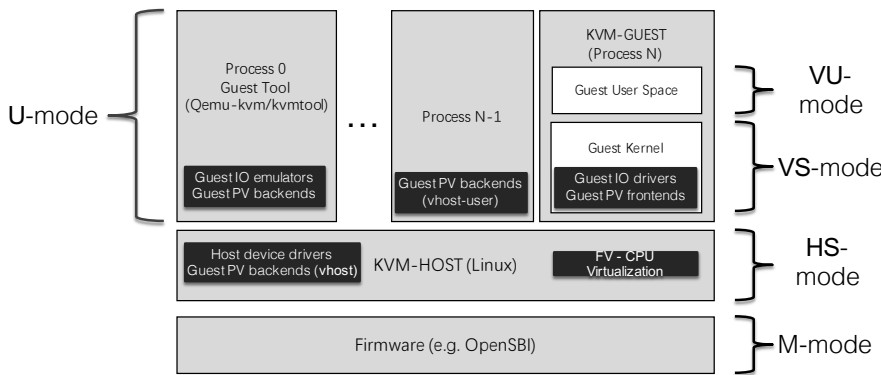

**Figure 4.** KVM on RISC-V.

Figure 5 shows the KHV based on RISC-V, in which KHV-ROOT is the root OS for the whole SoC system. We implemented the tool responsible for system virtualization settings, guest machine configuration, and startup OS. KHV-ROOT manages and isolates the system memory and KHV-BM/VM-GUEST. Then, KHV-ROOT is no longer the base hypervisor. Once KHV-GUEST starts up, it has equal privilege with KHV-ROOT (KHV cannot force control guests like KVM) and only uses the PV services provided by KHV-ROOT.

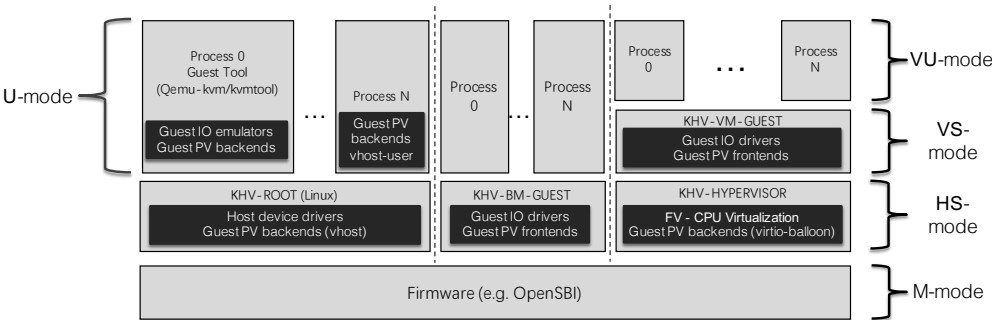

**Figure 5.** KHV on RISC-V.

Moreover, KHV-ROOT creates KHV-BM-GUEST without FV (CPU Full Virtualization) function and KHV-VM-GUEST with FV of KHV-HYPERVISOR. KHV-VM-GUEST has equal privilege with KHV-ROOT and only uses the PV service provided by KHV-ROOT. KHV-HYPERVISOR enables the upper-layer KHV-VM-GUESTs to use memory more efficiently and share memory through the two-level address translation of FV. The following introduces the internal designs such as KHV Memory Protection, KHV-HYPERVISOR, KHV virtio-balloon, Hotplug for CPU sharing, and KHV IO Emulation.

### 3.1. KHV Memory Isolation

In the KVM virtualization memory model, KVM-HOST can access the memory of KVM-GUEST, but KVM-GUEST cannot access the memory of KVM-HOST. KHV still follows this model, using the PMP function of the RISC-V ISA to restrict KHV-GUEST's access to memory. Figure 6 shows RISC-V SoC's memory access topology between PMP, IOPMP, and CPU peripherals.

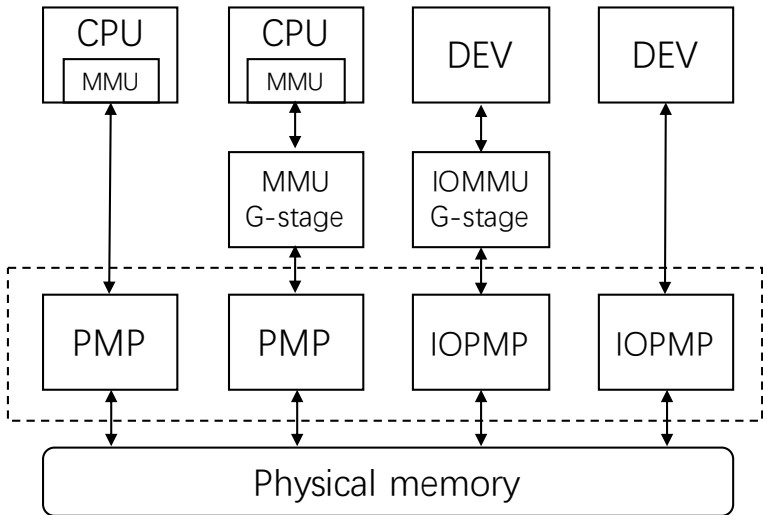

**Figure 6.** PMP and IOPMP in RISC-V SoC.

KHV-HYPERVISOR further utilizes the two-level address translation function of MMU and IOMMU to manage the memory access of KHV-VM-GUEST in a fine-grained manner.

### 3.2. KHV-HYPERVISOR

This paper innovatively designs a hypervisor shim (KHV-HYPERVISOR) to create KHV-VM-GUEST. KHV-HYPERVISOR uses the RISC-V MMU G-stage address translation mechanism to enhance the memory usage efficiency of the entire KHV system. KHV-HYPERVISOR is a unique form of KHV-BM-GUEST. KHV-HYPERVISOR follows the static-partitioning role, which does not occupy the CPU but supports KHV-VM-GUESTs' running. Each virtual CPU is bound to a physical CPU and uses the hot-plug method (the same with KHV-BM-GUEST) to manage and share CPUs. From the perspective of CPU management, there is no difference between KHV-VM-GUEST and KHV-BM-GUEST. However, from a memory management perspective, KHV-HYPERVISOR uses RISC-V virtualized MMU/IOMMU G-stage translation to achieve page-level memory isolation and sharing. The KHV-VM-GUESTs of KHV-HYPERVISOR can use non-contiguous physical memory to improve memory usage efficiency in KHV-HYPERVISOR. Figure 7 shows the implementation of KHV-HYPERVISOR using MMU/IOMMU G-stage functionality in a RISC-V SoC. KHV-ROOT and KHV-BM-GUEST directly bypass the G-stage MMU/IOMMU function only with PMP and IOPMP.

KHV-HYPERVISOR can reduce the consumption of contiguous physical memory. We introduce how to implement local memory sharing between KHV-VM-GUESTs in the next chapter of KHV virtio-balloon.

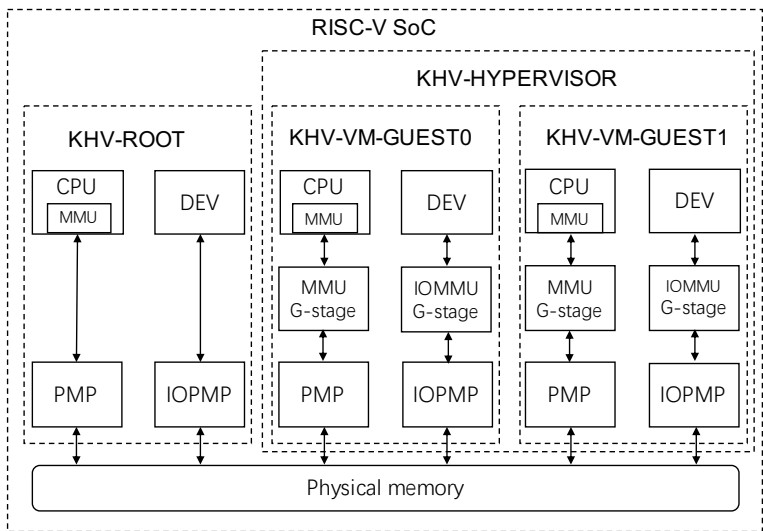

**Figure 7.** KHV-HYPERVISOR in a SoC.

### 3.3. KHV Virtio-Balloon

KVM's virtio-balloon function is popular in data center virtualization, letting KVM-HOST reclaim KVM-GUEST's memory by negotiation.

Figure 8 shows the difference in how virtio-balloon works on KVM and KHV systems. When KVM-GUEST returns the memory to KVM-HOST, the virtio-balloon front-end driver of KVM-GUEST will inflate and occupy more pages and then report the occupied memory to KVM-HOST. The KHV-HOST balloon backend driver unmaps these memory pages and recycles them into the free memory pool of KVM-HOST, which other guests or processes would reuse. KHV reuses the front-end driver of KVM but re-implements a new backend driver in KHV-HYPERVISOR, which only serves the KHV-VM-GUESTs in the current KHV-HYPERVISOR cluster. KHV-VM-GUEST will let the virtio-balloon frontend drive inflate when there is remaining memory and let KHV-HYPERVISOR reclaim the memory. When other KHV-VM-GUESTs have a memory requirement, the virtio-balloon frontend driver deflates and applies more memory from KHV-HYPERVISOR. In this way, the memory freed to KHV-HYPERVISOR can be flexibly shared among KHV-VM-GUESTs to improve the memory usage efficiency of the whole KHV system.

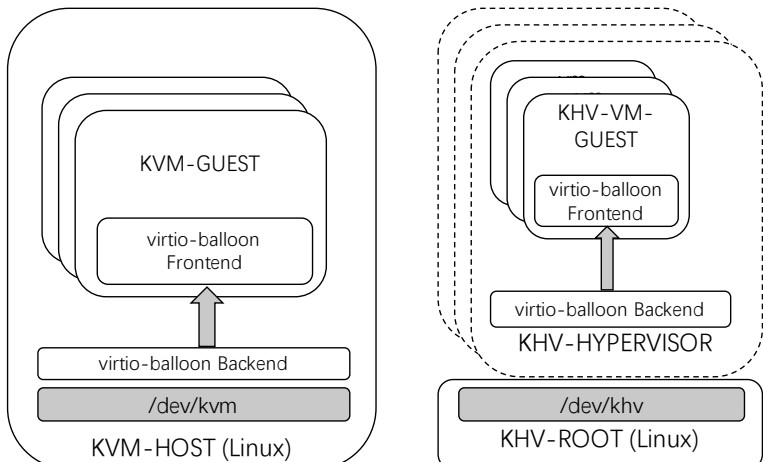

**Figure 8.** Virtio-balloon (KVM vs. KHV).

### 3.4. Hotplug For CPU Sharing

The CPU sharing method of KHV is different from that of KVM. It follows the principle of Static Partitioning by the hot-plug methodology. This method is also called CPU

pass-through mode. Other SoC hardware resources can also use this method of sharing. KVM uses Linux threads to simulate vCPUs, which causes the client to be affected by the Linux scheduler, resulting in performance jitter and the inability to guarantee real-time performance. KHV directly uses a physical CPU, so there is no such problem. Assuming that the system has N physical CPUs and requires M KHV guests (N > M), KHV will share the N-M CPUs between KHV-GUESTs. In this way, KHV achieves the maximum CPU utilization efficiency.

As shown in Figure 9, KHV implements the CPU hot-plug sharing mechanism based on OpenSBI. When the system started, all CPUs existed in KHV-ROOT. When KHV creates KHV-BM/VM-GUEST, CPUs will be unplugged from KHV-ROOT and plugged into KHV-GUEST. KHV-GUESTs have the same possible CPU slots to accommodate maximum CPU numbers. When the KHV-GUEST load is low, it will free its CPUs to the offline pool. When the KHV-GUEST load is high, it will try to request the CPU plug-in through OpenSBI's hot-plug API to meet the requirements of computing.

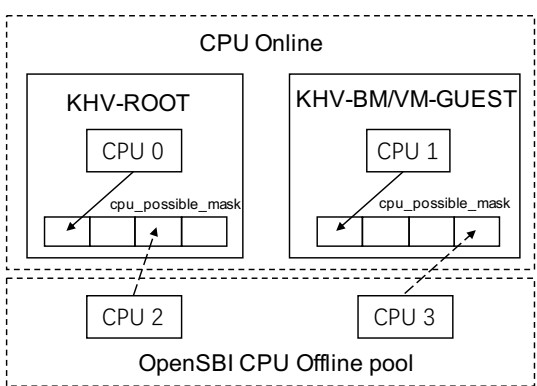

**Figure 9.** CPU online and offline diagram (KVM vs. KHV).

*3.5. KHV IO Emulation*

As shown in Figure 10, KVM IO Emulation is implemented based on CPU hardware virtualization. KHV creates a virtual IO address range (e.g., MMIO) by G-stage address translation for the guest. When the GUEST accesses the virtual IO address, it will trap into the KHV-ROOT backend driver, and then VM Exit happens and notifies the relevant processing thread by the eventfd. When the IO emulation thread finishes the GUEST trap request, it returns to the KHV-GUEST trap point with the VM Enter context switch.

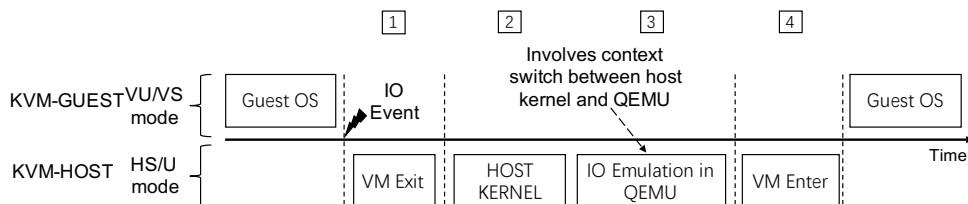

**Figure 10.** KVM IO emulation process.

KHV does not rely on CPU hardware virtualization and uses an "interrupt notification" and "shared memory" mechanism to implement KHV IO-emulation. It encapsulates a set of KHV-IO APIs for GUEST and solves spin–wait blocking problems with a FIFO queue design. As shown in Figure 11, the guest initiates an IO access interrupt request to the host through the KHV-IO API and enters the spin–wait state. After the KHV-ROOT driver receives the interrupt, it will use the eventfd mechanism to notify the relevant processing threads. Depending on the backend driver implementation, these working threads may be in user or kernel mode. After the IO Emulation backend driver thread completes the task, it returns to the KHV host driver and releases the guest spin–waiting state so the guest can continue executing. Therefore, the typical KHV-IO access implementation has

the problem of spin–wait blocking. There is no impact in the negotiation scenarios between frontend and backend devices. However, the virtio notify scenario will cause performance degradation. Therefore, we introduce a FIFO queue to solve the problem.

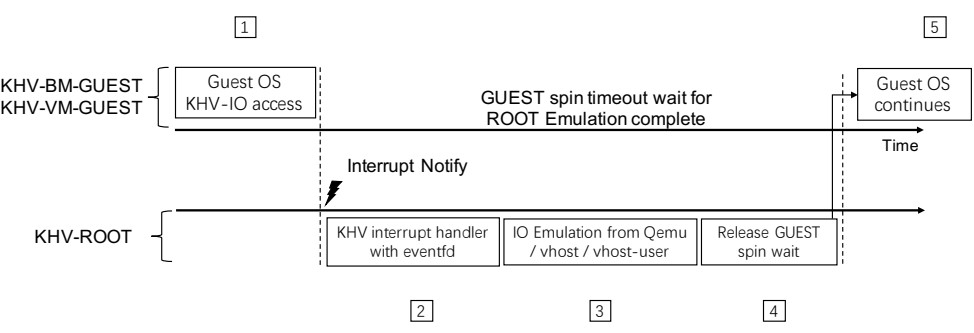

**Figure 11.** KHV IO emulation process.

As shown in Figure 12, notify is a special KHV-IO write operation. When KHV-GUEST writes the NOTIFY IO-emulation address, it will insert the request into the free slot of the FIFO queue. If the FIFO queue is empty, GUEST will raise an interrupt notification to KHV-ROOT. If the FIFO queue is non-empty, KHV-ROOT is still polling the FIFO ring and the GUEST need not raise interrupt. When the FIFO queue is full (no free slot for GUEST inserts), the GUEST will enter the spin–waiting state. The KHV-GUEST would block normal KHV-IO emulation read and write until the "Notify FIFO queue" is empty. So, KHV-IO transactions have a strong consistency.

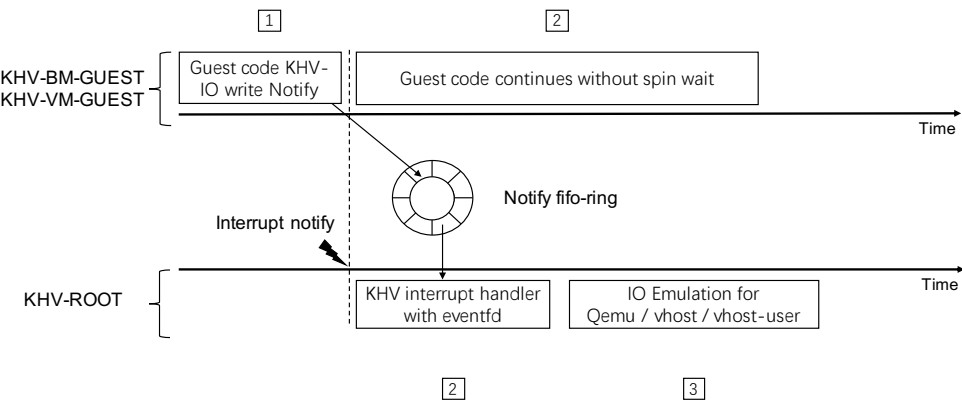

**Figure 12.** KHV notify IO emulation process.

### 3.6. Summary of Design

KHV follows the principle of static partitioning and does not rely on CPU hardware virtualization, which is the most significant difference from KVM. KHV has better guest isolation than KVM by "Memory Isolation with PMP" and has the same efficiency as KVM by memory and CPU resources sharing by "khv-hypervisor", "virtio-balloon", and "CPU hot-plug". Moreover, in the IO emulation, KHV naturally inherits KVM's para-virtualization mechanism and makes virtio-blk/net/vsock/console/rng/balloon run on the KHV system.

### 4. Evaluation

#### 4.1. Methodology Clarification

Since the RISC-V Xuantie 910 CPU does not support hardware-assisted virtualization, it is impossible to directly compare the performance differences between KHV and KVM on the same platform. Therefore, we chose the SoC platform of RISC-V Xuantie 910 CPU and the SoC platform (Raspberry Pi 4b) of ARM Cortex-A72 CPU for comparative measurement

as they have similar performance. The difference between the virtual machine running system and the native running system was used to compare KHV and KVM in real-time performance and stability (i.e., Difference Comparison Method):

Obtain the test result 1 on KHV-BM-GUEST and result 2 on KHV-ROOT for the same test item A, then 'result 1/result 2' is the ratio of KHV-BM-GUEST to KHV-ROOT. For example, if the ratio is 0.95, it means that KHV-BM-GUEST can only achieve 95 percent of the performance of KHV-ROOT for test item A. Similarly, obtain the test result result 3 on KVM-GUEST and result 4 on KVM-HOST, then 'result 3/result 4' is the ratio of KVM-GUEST to KVM-HOST. For example, if the ratio is 0.85, it means that KVM-GUEST can only achieve 85 percent of the performance of KVM-HOST; thus, we conclude that KHV outperforms KVM (0.95 > 0.85) for test item A. Table 2 lists the platforms' parameters of the experiment.

**Table 2.** Introduction to the test platform.

|  | **KHV: RISC-V XuanTie 910 SoC Platform** | **KVM: ARM Cortex-A72 SoC Platform** |
|---|---|---|
| CPU | T-HEAD XuanTie 910 $\times$ 4 | ARM Cortex-A72 $\times$ 4 |
| CPU freq | 1.8 Ghz | 1.5 Ghz |
| L2 cache | 1 M | 1 M |
| L1 icache | 64 KB | 48 KB |
| L1 dcache | 64 KB | 32 KB |
| RAM | 4 G LPDDR4X-3733 | 4 GB LPDDR4-2400 |
| GUEST | 2 harts, 1 GB RAM | 2 harts, 1 GB RAM |
| User tool | kvmtool (khv-backend) | kvmtool (arm-backend) |

*4.2. Context Switch Comparison*

In this section, we compare the OS scheduling performance of KHV and KVM with Hackbench and Lmbench (context switch). The configurations of the test items are shown in Table 3.

**Table 3.** Hackbench and Lmbench context switch.

|  | **Hackbench** | **Lmbench Context Switch** |
|---|---|---|
| Function description | Use Hackbench to create 750 pairs of reading and writing tasks. Each writing task sends 200 times $\times$ 512 bytes of data to the reading task through the pipeline, the reading task reads the data and finally checks the execution time of the test program (the shorter the time, the better) | Use Lmbench (context switch) to create a test environment with a total of 8 parent and child processes. The amount of data exchanged when switching between parent and child processes is 16 Kbytes. Finally, check the execution time of the test program (the shorter the time, the better) |
| Test cmd | hackbench -s 512 -l 200 -g 15 -f 25-p″ | lat_ctx -s 16 8 |

We tested Hackbench and Lmbench (context switch) 50 times, respectively, plotted the results of the 50 tests into a graph, and obtained the variance of the test data (variance is used to measure the degree of deviation between a set of sample values and their mean, it is the mean of the squared values of the difference between each sample value and the mean). As shown in Figure 13, the curve marked KHV represents the test result on KHV-BM-GUEST, and the curve marked KVM represents the test result on KVM-GUEST. As shown on the left, the variance of the KHV curve (0.022) in the Hackbench test is smaller than the variance of the KVM curve (0.060); as shown on the right, the variance of the KHV curve (0.392) in the Lmbench (context switch) test is smaller than that of the KVM curve (0.960). The experimental shows that KHV is 50% smaller than KVM in terms of fluctuation. So, KHV has less impact on the scheduling performance of the guest OS and better real-time performance and stability than KVM.

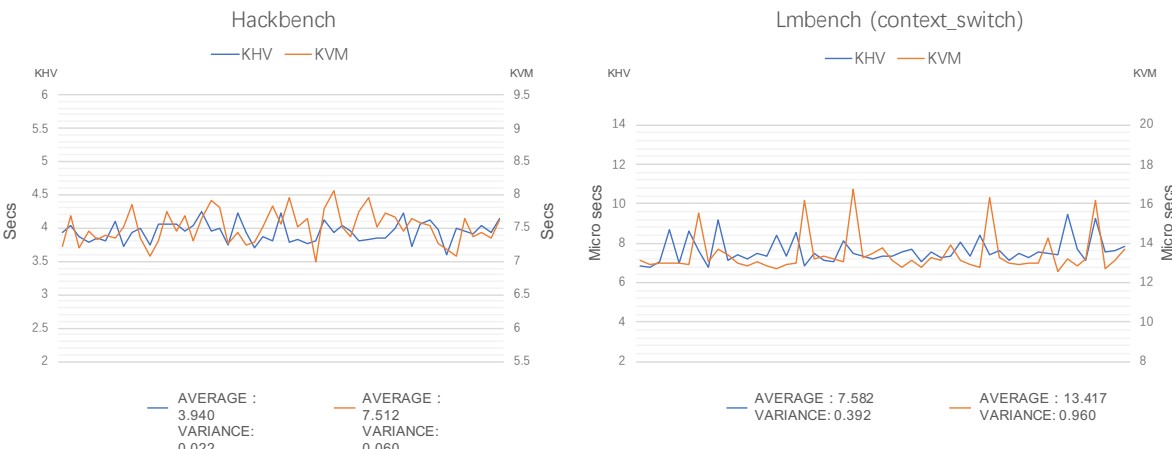

**Figure 13.** Context switch performance fluctuation graph.

Similarly, use Hackbench and Lmbench (context switch) to compare the GUEST performance drop of KHV and KVM. As shown in Figure 14, the value marked as KHV in the figure represents the ratio of KHV-BM-GUEST to KHV-ROOT, and the value marked as KVM represents the ratio of KVM-GUEST to KVM-HOST. Regardless of Hackbench or Lmbench (context switch), KHV-BM-GUEST performance is on par with KHV-ROOT. However, KVM-GUEST performance can only reach 86 percent (Hackbench) and 58 percent (Lmbench) of KVM-HOST. The experiment shows that KHV lets the guest OS have the same performance with bare-metal OS, whereas KVM dropped an average of 28%. So, KHV is significantly better than KVM in guest OS isolation.

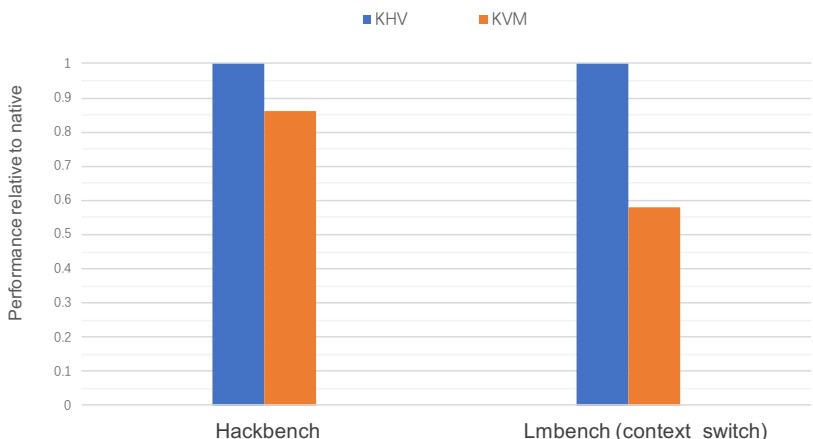

**Figure 14.** Performance KHV-BM-GUEST to KHV-ROOT (for KVM, KVM-GUEST to KVM-HOST).

*4.3. Benchmark Comparison*

We conducted a comparison test of the difference between KHV and KVM virtualization technologies by multiple popular benchmarks. The results in Figure 15 show that the performance of KHV-BM-GUEST has no performance impact compared with KHV-ROOT, but KVM has a small performance drop. This is because KHV does not rely on hardware virtualization technology, whereas KVM-GUEST requires hardware virtualization, resulting in a virtualization tax and slightly lower performance than KVM-HOST.

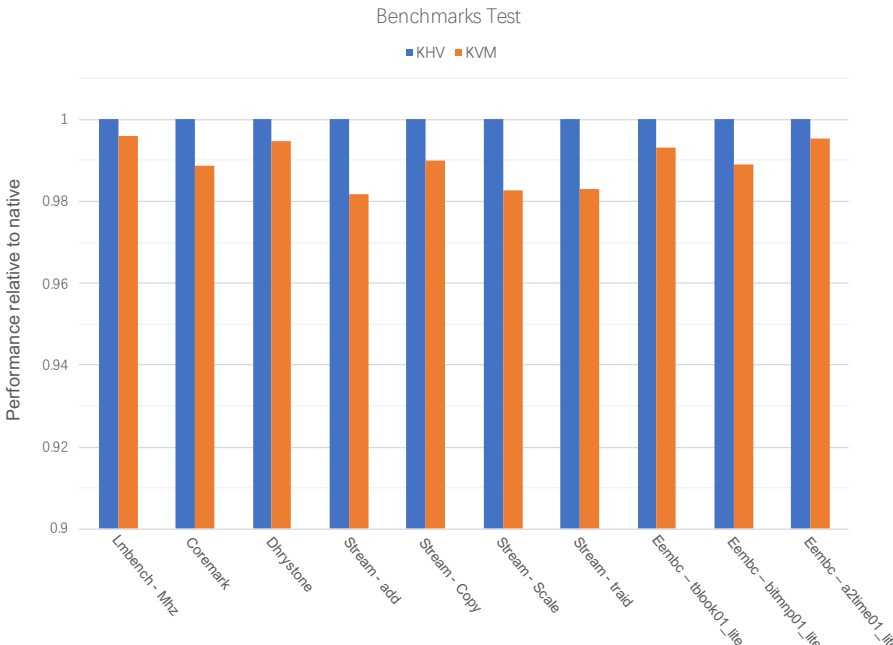

**Figure 15.** Benchmark test results.

### 4.4. Virtio Performance Comparison

We compared the impact of KHV and KVM on the throughput and IO-delay performance of virtio-blk by SD-card and Ramdisk. Testing with an SD card as the backend of virtio-blk can more realistically reflect the impact of KHV and KVM in actual application scenarios. Testing with Ramdisk as the backend of virtio-blk can eliminate the effect of slow physical block devices.

We used the same SD card as the Virtio-blk backend storage device of KHV-BM-GUEST and KVM-GUEST. The experimental involves 4 kinds of operations (read, write, randread, and randwrite). The test results Figure 16 shows that the IO delay and the throughput of KHV-BM-GUEST are closer to KHV-ROOT, especially the performance of read and randread is significantly better than that of KVM-GUEST. From the throughput view, both KHV-BM-GUEST and KVM-GUEST can achieve the expected performance of bare metal because the bottleneck is on the SD card.

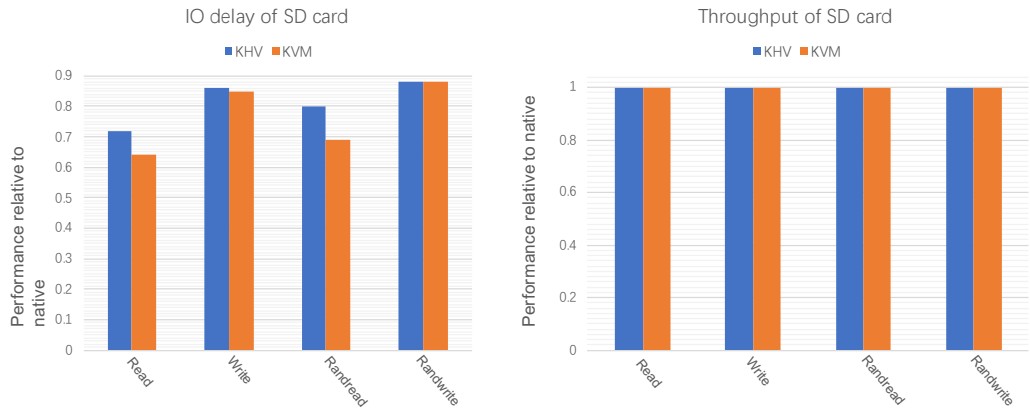

**Figure 16.** IO delay and throughput of SD card.

Since the SD card is the bottleneck of performance, the performance of KHV is the same as KVM. Therefore, we use ramdisk as the virtio-blk backend storage device. The test results in Figure 17 show that KHV outperforms KVM.

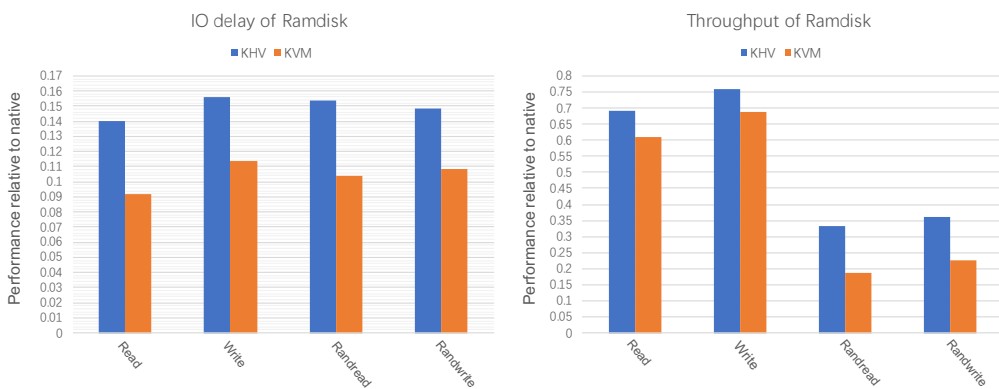

**Figure 17.** IO delay and throughput of ramdisk.

## 5. Discussion

### 5.1. The Critical Aspects of the Results

The experimental result shows that the performance of KHV-BM-GUEST is not different from that of KHV-ROOT. However, KVM turning on hardware virtualization drops the hackbench and Lmbench context switch performance. Additionally, the experimental result shows that KHV is smaller than KVM in terms of fluctuation, which proves the isolation of KHV is better because KHV does not depend on hardware virtualization and eliminates the tax of virtualization.

### 5.2. Shared Resource Contention

The current hypervisors based on Static Partition all have the problem of guest competition for system cache and memory bandwidth in one SoC. Because Static Partition only solves the problem of address space isolation, the CPU and peripherals in the SoC still share the memory bandwidth and system cache, leading to bouncing or low memory bandwidth. So, it would cause a performance jitter, negatively affecting critical tasks' real time and reliability. This problem is called "noisy neighbor", a known problem of a traditional static partitioning hypervisor. KHV also has this problem, and some solutions have already appeared (e.g., Bao Hypervisor [3] partitions the last level cache of the CPU based on the cache coloring mechanism with the G-stage address translation of CPU virtualization). Bao's method also has apparent disadvantages:

- It relies on the G-stage translation of CPU virtualization;
- The problem of memory bandwidth allocation cannot be solved;
- Unable to use hugepage;
- Divides the physical memory into pieces.

There are better QoS solutions than Bao's, such as Intel's RDT technology [20], which allocates and prioritizes system cache and memory bandwidth through CAT and MBA. Intel's RDT can satisfy processors, virtual machines, containers, applications, cache isolation, and memory bandwidth allocation requirements to ensure enough system resources for critical tasks. In the future, we will research the combination of KHV and QoS tech to solve problems such as shared resource contention.

## 6. Conclusions

The innovation of KHV is to implement a static partitioning hypervisor independent of virtualization and an optional localized hypervisor. KHV aims to bring the KVM software ecosystem from the data center to the embedded field and meet the requirements of embedded reliability and security. KHV follows the principle of static partitioning, uses the hypervisor locally, and inherits the KVM software ecosystem. KHV balances the demands of static partitioning and flexible sharing in the embedded system. The experiment shows that KHV is 50% smaller than KVM in terms of fluctuation, and KHV makes KHV-BM-GUEST achieve the same performance as the bare-metal OS in scheduler benchmarks,

whereas KVM dropped an average of 28%. RISC-V is still a blank piece of paper in the field of embedded virtualization, and few companies have entered the RISC-V embedded virtualization market. KHV does not rely on hardware virtualization technology, and it meets the new opportunity to lead the development of RISC-V architecture in the field of embedded virtualization.

**Author Contributions:** Conceptualization, C.L. and R.G.; methodology, C.L. and R.G.; software, C.L., R.G. and X.T.; validation, H.W.; project administration, C.L.; funding acquisition, C.L. All authors have read and agreed to the published version of the manuscript.

**Funding:** This research received no external funding.

**Institutional Review Board Statement:** Not applicable.

**Informed Consent Statement:** Informed consent was obtained from all subjects involved in the study.

**Data Availability Statement:** Not applicable.

**Acknowledgments:** Patel, A.; Daftedar, M.; Shalan, M.; El-Kharashi, M.W. Embedded hypervisor xvisor: A comparative analysis, and Zhang, X.; Zheng, X.; Wang, Z.; Yang, H.; Shen, Y.; Long, X. High-density multi-tenant bare-metal cloud.

**Conflicts of Interest:** The authors declare no conflict of interest.

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
