# Peer review of "KHV: KVM-Based Heterogeneous Virtualization"

_electronics, doi:10.3390/electronics11162631_

Round 1
Reviewer 1 Report
The organization of the manuscript is not correct. Related Work should be after introduction. See good work in the domain 10.1007/s11277-018-5376-3, 10.1007/s40009-022-01112-y
Improve the related work with actual research gaps and problem.
The critical aspects of the results are not discussed. Revise the conclusion accordingly
Author Response
Very grateful to receive the reviewer's comments.
As the example papers do, we will improve the related work by emphasizing the research gap and putting it right after the Introduction section in our revised version.
Besides, we are adding some discussion about the evaluation results.
We believe these revisions based on your suggestions will improve our paper.
Reviewer 2 Report
This paper proposes a KVM-based Heterogeneous Virtualization (KHV), which follows 3 the principle of static partitioning, localizes the hypervisor, and inherits the KVM software ecosystem. 4 KHV balances the demands of static partitioning and flexible sharing in the embedded system. This topic is up-to-date. However, the novelty and the contribution of this paper are good. The paper is well written and the analyses along with the results are well expressed. These are some of the comments that need to be addressed.
1. The presentation contains several grammar errors; English must be improved too.
2. Abstract: should highlight the main idea of the KVM-based Heterogeneous Virtualization (KHV); specifically, what is novel.
3. Introduction: needs a stronger motivation.
4. There have been a lot of papers published on the a KVM-based Heterogeneous Virtualization. Consider recent papers also.
5. Why KVM consider RISC V? What is the difference in ARM design? Consider following link and mention your proposed work.
http://www.virtualopensystems.com/en/solutions/guides/kvm-on-arm/
6. Explain Fig. 7 clearly?
7. What is the reason PMP function of the RISC-V ISA to restrict KHV-GUEST’s access to memory?
8. Why consider for comparison KHV only? Consider recent works also.
9. References are not format (For example: Ref 9, Ref 10 …)
Author Response
Very grateful to receive your comments. Here are the replies.
> 1. The presentation contains several grammar errors; English must be improved too.
A: Em..., we will recheck the English grammar in the next revision.
> 2. Abstract: should highlight the main idea of the KVM-based Heterogeneous Virtualization (KHV); specifically, what is novel.
A: We innovate a new hypervisor named KHV. KHV != KVM, but based on KVM, reuse the paravirt ecosystem of KVM. KHV does not enforce CPU hardware virtualization, but KVM must enforce CPU hardware virtualization. Although users use the same kvmtool and it is difficult to tell whether the underlying layer is KHV or KVM, KHV and KVM are different technologies.
> 3. Introduction: needs a stronger motivation.
A: Okay, We will emphasize motivation in the next revision.
> 4. There have been a lot of papers published on the a KVM-based Heterogeneous Virtualization. Consider recent papers also.
We have done extensive research, and there is no paper similar to KHV before this paper. The KHV adds a new directory linux/virt/khv/ in the Linux kernel source tree, which is side by side with linux/virt/kvm/.
KHV does not enforce CPU hardware virtualization.
KVM must enforce CPU hardware virtualization.
The most related paper to KHV is:
18. High-density multi-tenant bare-metal cloud
19. Nohype: virtualized cloud infrastructure without the virtualization
> 5. Why KVM consider RISC V? What is the difference in ARM design? Consider following link and mention your proposed work.
Not KVM; we proposed KHV. The article you gave is about arm KVM design, not related to KHV. RISC-V also has implemented KVM, ref the link: https://www.phoronix.com/news/Linux-5.16-KVM-RISC-V
> 6. Explain Fig. 7 clearly?
Fig. 7 shows you how KHV-ROOT and KHV-VM-GUEST are placed in one SoC. KHV-ROOT & KHV-BM-GUEST needn't hardware virtualization, which means they needn't MMU G-stage translation (called by RISC-V, Intel calls it level-2 translation, Arm calls it stage-2 translation). But KHV-VM-GUEST could utilize hardware virtualization to enhance memory utilization in some low-safety scenarios. Thus, they enable MMU G-stage translation in the KHV-HYPERVISOR domain.
> 7. What is the reason PMP function of the RISC-V ISA to restrict KHV-GUEST’s access to memory?
KHV-ROOT & KHV-BM-GUEST didn't enable hardware virtualization, so we need PMP (Physical Memory Protection) to deal with the phyical address.
ARM doesn't have PMP, so we can't implement KHV on the Arm by default ISA. (Maybe also answer the question 6)
> 8. Why consider for comparison KHV only? Consider recent works also.
We compared the KHV with KVM and showed that KHV has a better effect on safety isolation.
> 9. References are not format (For example: Ref 9, Ref 10 …)
We use CVE public ref format, which founded in a lot of papers:
@misc{CVE-2007-4993,
title = {{CVE}-2007-4993.},
howpublished = "Available from MITRE, {CVE-ID} {CVE}-2007-4993.",
url={http://cve.mitre.org/cgi-bin/cvename.cgi?name=CVE-2007-4993},
}
What's the format which you think is right? Thx.
Reviewer 3 Report
The paper is well written and the experimental results shows the usefulness
of the proposed approach.
There are a couple of things to be better to be mentioned which are
shown in the attached paper as comments.

Author Response
Very grateful to receive your comments.
I've put my replies in the pdf attach.
Best Regards
Guo Ren

Round 2
Reviewer 1 Report
Suggested changes are done by the authors